# Male germ cells support long-term propagation of Zika virus

Christopher L. Robinson[1], Angie C. N. Chong[1], Alison W. Ashbrook[2], Ginnie Jeng[1], Julia Jin[1], Haiqi Chen[3], Elizabeth I. Tang[3], Laura A. Martin[1], Rosa S. Kim[4], Reyn M. Kenyon[1], Eileen Do[1], Joseph M. Luna[2], Mohsan Saeed[2], Lori Zeltser[5,6], Harold Ralph[7], Vanessa L. Dudley[8], Marc Goldstein[9], Charles M. Rice[2], C. Yan Cheng[3], Marco Seandel[1] & Shuibing Chen[1,10]

Evidence of male-to-female sexual transmission of Zika virus (ZIKV) and viral RNA in semen and sperm months after infection supports a potential role for testicular cells in ZIKV propagation. Here, we demonstrate that germ cells (GCs) are most susceptible to ZIKV. We found that only GCs infected by ZIKV, but not those infected by dengue virus and yellow fever virus, produce high levels of infectious virus. This observation coincides with decreased expression of interferon-stimulated gene *Ifi44l* in ZIKV-infected GCs, and overexpression of *Ifi44l* results in reduced ZIKV production. Using primary human testicular tissue, we demonstrate that human GCs are also permissive for ZIKV infection and production. Finally, we identified berberine chloride as a potent inhibitor of ZIKV infection in both murine and human testes. Together, these studies identify a potential cellular source for propagation of ZIKV in testes and a candidate drug for preventing sexual transmission of ZIKV.

[1] Department of Surgery, Weill Cornell Medical College, 1300 York Avenue, New York, NY 10065, USA. [2] Laboratory of Virology and Infectious Disease, Center for the Study of Hepatitis C, The Rockefeller University, New York, NY 10065, USA. [3] The Mary M. Wohlford Laboratory for Male Contraceptive Research, Center for Biomedical Research, Population Council, 1230 York Avenue, New York, NY 10065, USA. [4] Penn State College of Medicine, Hershey PA 17033, USA. [5] Naomi Berrie Diabetes Center, Columbia University, New York, NY 10032, USA. [6] Department of Pathology and Cell Biology, Columbia University, New York, NY 10032, USA. [7] Weill Cornell Medical College-Microscopy and Image Analysis Core Facility, 1300 York Avenue, New York NY 10065, USA. [8] Institute of Reproductive Medicine at Weill Cornell Medicine, Weill Cornell Medicine–New York Presbyterian Hospital, New York, NY 10065, USA. [9] Department of Urology and Institute for Reproductive Medicine, Weill Cornell Medical College of Cornell University, New York, NY 10065, USA. [10] Department of Biochemistry, Weill Cornell Medical College, 1300 York Avenue, New York, NY 10065, USA. These authors contributed equally: Christopher L. Robinson, Angie C.N. Chong. Correspondence and requests for materials should be addressed to C.Y.C. (email: y-Cheng@popcbr.rockefeller.edu) or to M.S. (email: mas9066@med.cornell.edu) or to S.C. (email: shc2034@med.cornell.edu)

Male-to-female sexual transmission of Zika virus (ZIKV), as seen in recent outbreaks, revealed an unexpected mode of transmission for a viral infection once thought to be transmitted primarily by *Aedes aegypti* mosquitoes[1, 2]. The presence of ZIKV in human semen[3–5] and sperm[6] up to 6 months after infection, along with the absence of ZIKV in the peripheral circulation, suggests a potential role for testicular cells in the propagation of ZIKV. Immunocompromised murine models of ZIKV infection implicate the proximal male reproductive tract (i.e., testis and epididymis) as the target of ZIKV infection, demonstrate catastrophic effects on the testis, and reveal that multiple cell types, including germ cells (GCs), Sertoli cells (SCs), Leydig cells (LCs), and testicular peritubular-myoid cells (MCs), are vulnerable to infection and destruction by ZIKV[7, 8]. Although no studies to date have reported ZIKV-induced acute orchitis in humans, the effects of ZIKV in immunocompetent men are more subtle and potentially amenable to therapeutic targeting. While various cell types are susceptible to ZIKV infection in interferon (IFN) receptor 1-deficient mice (*Ifnar1*[−/−])[7–9], ZIKV susceptibility in the testes of wild-type immunocompetent mice has not been systematically evaluated. Furthermore, it remains unknown which testicular cells are capable of sustaining production of infectious ZIKV. In this study, we investigated the susceptibility of different cell types in murine testicular tissue to ZIKV infection, the ability of these testicular cells to produce infectious ZIKV, and the application of a small molecule to inhibit infection and production of ZIKV in both murine and human testis.

## Results

**ZIKV infection of male murine GCs.** To determine whether ZIKV infection can persist in wild-type mice, we infected immunocompetent wild-type mice with $1 \times 10^8$ infectious units (IFUs) ZIKV (strain: MR 766) by intravenous (IV) injection and measured ZIKV titers in the serum and in various organs. ZIKV viral RNA (vRNA) was detected in the serum of all mice at 2 days post-infection (dpi), and, by 14 dpi, serum ZIKV vRNA levels in infected mice had returned to baseline presumably due to host immunity (Fig. 1a). ZIKV vRNA was detected in 100% of testes at 7 dpi, 54% at 14 dpi, and 30% at 30 dpi (Fig. 1b). At 60 dpi, 14% of the testes were ZIKV positive (Fig. 1c). We also observed a decrease in sperm count in mice positive for ZIKV vRNA in their testis as indicated by real-time quantitative reverse transcription PCR (qRT-PCR) (Fig. 1d). Only one testis in the cohort of wild-type mice was atrophic (Fig. 1e), as seen in immunocompromised *Ifnar1*[−/−] mice[7, 8], and exhibited morphological changes by hematoxylin and eosin (H&E) staining (Fig. 1f) at 60 dpi. ZIKV vRNA was not detected in the serum, brain, prostate, and seminal vesicle of ZIKV-infected mice at 60 dpi (Supplementary

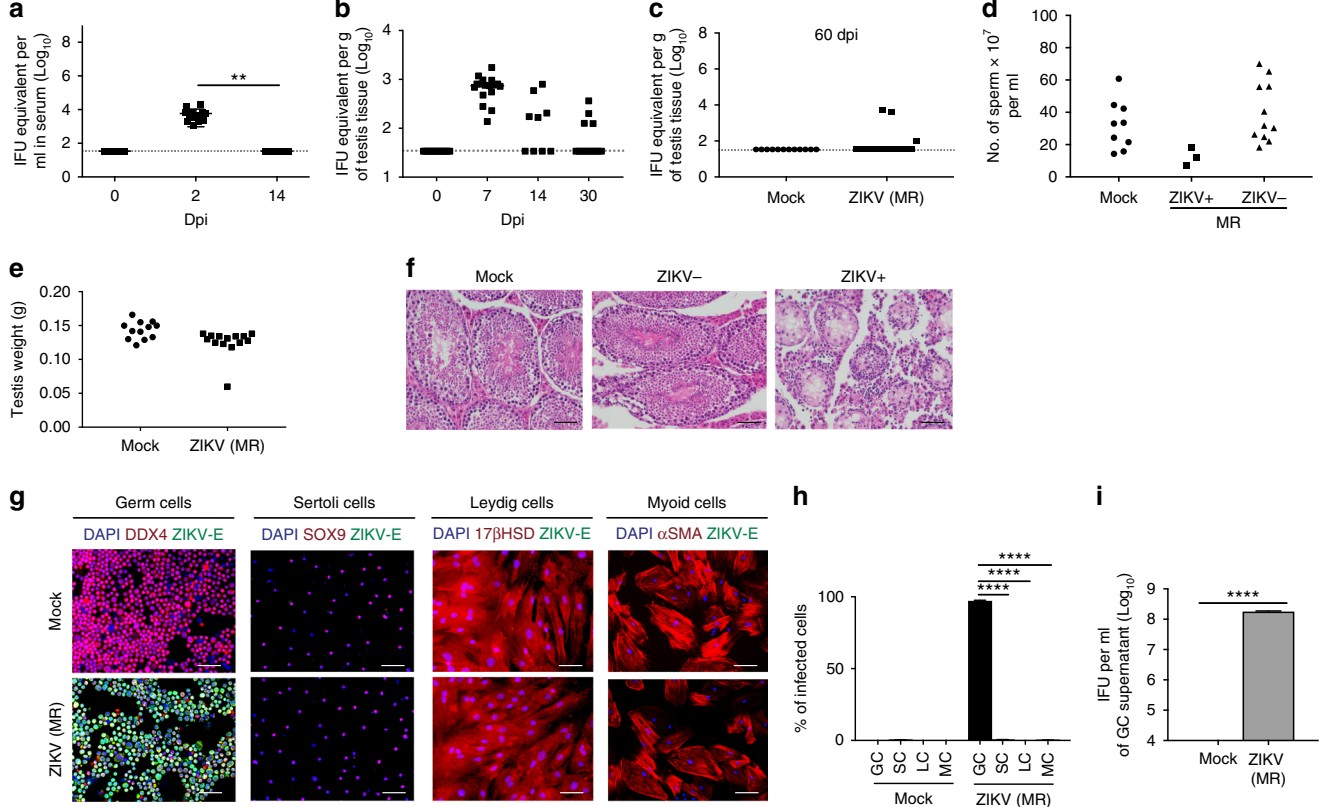

**Fig. 1** Murine male GCs exhibit increased susceptibility to ZIKV infection. **a–c** qRT-PCR analysis of ZIKV vRNA in ZIKV-infected (IFU = 1 × 10$^8$) wild-type CD-1 mice in serum at 0, 2, and 14 dpi ($n = 8$, 16, and 7, respectively) (**a**); testis at 0, 7, 14, and 30 dpi ($n = 13$, 17, 9, and 13, respectively) (**b**); and testis at 60 dpi ($n = 12$ for mock-infected and 14 for ZIKV-infected) (**c**). The dashed line represents the limit of detection. **d** Sperm count of mock-infected ($n = 10$) and ZIKV-infected (IFU = 1 × 10$^8$) with detectable (ZIKV+; $n = 3$) and undetectable (ZIKV−; $n = 11$) vRNA in the testis of wild-type CD-1 mice at 60 dpi. **e** Testis weight in mock-infected and ZIKV-infected (IFU = 1 × 10$^8$) wild-type CD-1 mice at 60 dpi ($n = 12$ for mock infected and 14 for ZIKV infected). **f** Hematoxylin and eosin staining of mock, ZIKV−, and ZIKV+ testis at 60 dpi. **g**, **h** Mock-infected and ZIKV-infected (MOI = 0.1 PFU per cell) GCs (DDX4), SCs (SOX9), LCs (17β-HSD), and MCs (α-SMA) immunostaining (**g**) and quantification ($n = 5$, 3, 5, and 8, respectively, for each group) (**h**). **i** Quantification of infectious ZIKV in supernatant of mock-infected and ZIKV-infected (MOI = 0.1 PFU per cell) GCs at 72 hpi by intracellular flow cytometry-based Vero assay ($n = 3$). Data are presented as mean ± s.e.m. and analyzed by two-sided $t$ test and one-way ANOVA, **$p \leq 0.01$, or ****$p \leq 0.0001$. Scale bar, 50 μm. ZIKV strain is MR 766

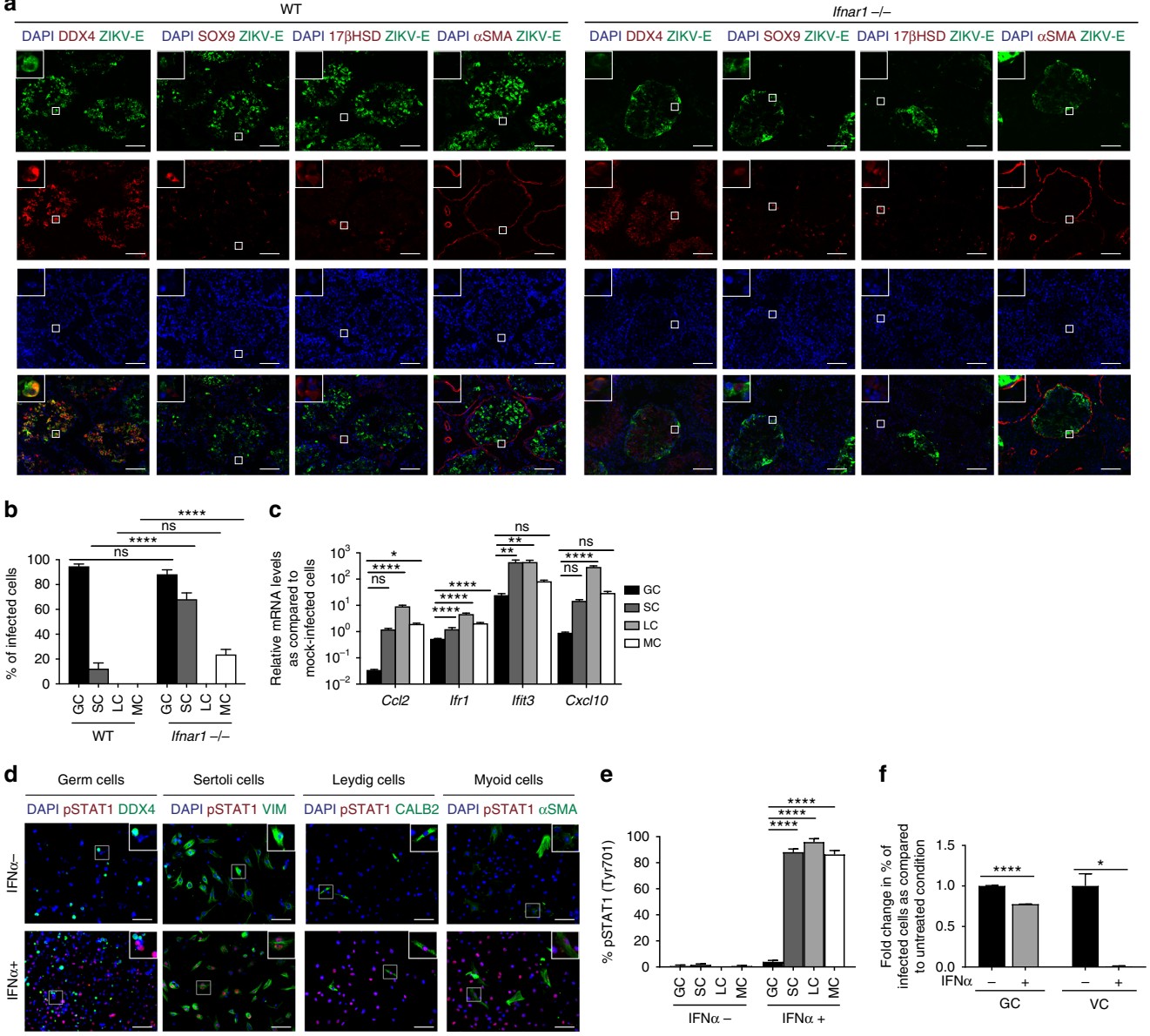

**Fig. 2** Increased susceptibility to ZIKV infection in murine male GCs due to reduced IFN signaling. **a**, **b** Immunostaining (**a**) and quantification (**b**) of ZIKV-infected wild-type C57/BL6J (IFU = 1 × 10⁵) and *Ifnar1⁻/⁻* (IFU = 1 × 10²) testis at 6 dpi. **c** Induction of ISGs in ZIKV-infected (MOI = 0.1 PFU per cell) GCs, SCs, LCs, and MCs at 72 hpi (*n* = 3) measured by qRT-PCR. **d**, **e** Immunostaining (VIM for SCs and CALB2 for LCs) (**d**) and quantification (*n* = 5 for each group) (**e**) of pSTAT1 (Tyr701) after 1 h of 1000 U ml⁻¹ of IFNα. **f** Percent of infected GCs and Vero cells (*n* = 3) measured by flow cytometry after a 24 h incubation with 1000 U ml⁻¹ of IFNα followed by a 1 h incubation with ZIKV (MOI = 0.1 PFU per cell). Data are presented as mean ± s.e.m. and analyzed by two-sided *t* test and one-way ANOVA, *$p \le 0.05$, **$p \le 0.01$, ****$p \le 0.0001$, or no significance (ns). Scale bar, 50 μm. ZIKV strain is MR 766

Fig. 1a–d), suggesting that testicular cells are selectively able to support continuous propagation of ZIKV in mice. To identify testicular cell types permissive for ZIKV, we infected murine testicular cells, GCs, SCs, LCs, and MCs, with ZIKV in vitro (MR 766, multiplicity of infection (MOI) = 0.1 plaque-forming unit (PFU) per cell), and stained for ZIKV envelope protein (ZIKV-E) and testicular cell markers, DEAD-Box helicase 4 (DDX4) for GCs, SRY-box 9 (SOX9) for SCs, 17β-hydroxysteroid dehydrogenases (17β-HSD) for LCs, and α-smooth muscle actin (α-SMA) for MCs. Cells costaining with ZIKV-E were 97.2% of DDX4⁺ cells, 0.53% of SOX9⁺ cells, 0.00% of 17β-HSD⁺ cells, and 0.21% of α-SMA⁺ cells (Fig. 1g, h). ZIKV-infected GCs generated 1.78 × 10⁸ IFU ml⁻¹ (Fig. 1i). Similar results were obtained when GCs, SCs, LCs, and MCs were infected with ZIKV strain PRVABC59 (Supplementary Fig. 1e–g), demonstrating that

susceptibility of GCs to ZIKV infection is a general property of even divergent ZIKV isolates. We further evaluated the susceptibility of GCs, SCs, LCs, and MCs to ZIKV infection in vivo. At 6 dpi, the testes of immunocompetent wild-type mice were co-stained for ZIKV-E and testicular cell markers (Fig. 2a). Consistent with the in vitro result (Fig. 1g, h), GCs showed increased ZIKV infection relative to SCs, LCs, and MCs (Fig. 2b). Colocalization with another marker DAZL (Deleted in Azoospermia Like), a spermatocyte marker[10], demonstrated that ZIKV mainly infected GCs, including the spermatocyte population (Supplementary Fig. 1h).

**Murine male GCs show reduced type 1 IFN signaling.** To determine the role of type 1 IFN signaling in ZIKV infection of GCs, we compared the susceptibility of GCs, SCs, LCs, and MCs in

wild-type and *Ifnar1⁻ʹ⁻* mice by immunostaining. We found that SCs and MCs in wild-type mice were significantly less susceptible to infection than the SCs and MCs in *Ifnar1⁻ʹ⁻* mice. In wild-type testis, only 12.3% of SOX9⁺ and 0.0% of α-SMA⁺ cells were co-stained with ZIKV-E. In *Ifnar1⁻ʹ⁻* testes, 68.2% of SOX9⁺ cells and 23.7% of α-SMA⁺ cells were co-stained with ZIKV-E, yet there was no difference in GC infection between the two models (Fig. 2a, b). To explore possible mechanisms that might contribute to enhanced GC susceptibility, we performed qRT-PCR analysis of mock-infected and ZIKV-infected GCs, SCs, LCs, and MCs. We found several IFN-stimulated genes (ISGs), including *Ccl2*, *Ifr1*, *Ifit3*, and *Cxcl10*, that were less upregulated in GCs compared with other testicular cell types, suggesting that IFN signaling is dampened in infected GCs (Fig. 2c). Since type I IFN signaling is important for limiting ZIKV infection[11], we investigated the response of cultured primary murine testicular cells to IFNα treatment by monitoring the levels of a downstream target, phosphorylated signal transducer and activator of transcript 1 (pSTAT1 (Tyr701)); 88.1% vimentin (VIM)⁺ SCs, 95.8% of calbindin 2 (CALB2)⁺ LCs, and 86.4% of α-SMA⁺ MCs stained positive for pSTAT1, whereas only 4.0% of DDX4⁺ GCs were positively stained (Fig. 2d, e). We next investigated whether IFNα could prevent ZIKV infection of GCs by pretreating GCs and Vero cells (VCs) with IFNα. While IFNα completely prevented ZIKV VC infection, GCs showed only a 23%

reduction in the percentage of infected cells when cultured in the presence of 1000 U ml⁻¹ IFNα (Fig. 2f). Together, these data suggest that reduced type I IFN signaling in GCs may be linked to their susceptibility to ZIKV infection, while active signaling may limit ZIKV infection of SCs and MCs.

**ZIKV infection does not influence cell proliferation or survival.** Studies have demonstrated that ZIKV impairs cell proliferation and triggers apoptosis of neural progenitor cells (NPCs) in culture and adult mouse brain[12, 13]; therefore, we examined the expression of Ki67, a cell proliferation marker, and cleaved caspase-3, an apoptotic cell marker, in GCs at 72 h post-infection (hpi) (MOI = 0.1 PFU per cell). In contrast to NPCs, no significant differences were detected in the expression levels of either Ki67 or cleaved caspase-3 in ZIKV-infected GCs as compared with mock-infected controls (Fig. 3a–d). ZIKV infection resulted in a modest but significant decrease in the percentage of cells in the 4N phase and an increase in cells in the 2N phase, with no change in the S phase population (Fig. 3e–h). Taken together, these findings indicate that ZIKV induces minor effects on the cell cycle, but no observable effect on cell proliferation or survival, consistent with the notion that infected GCs could maintain durable ZIKV production.

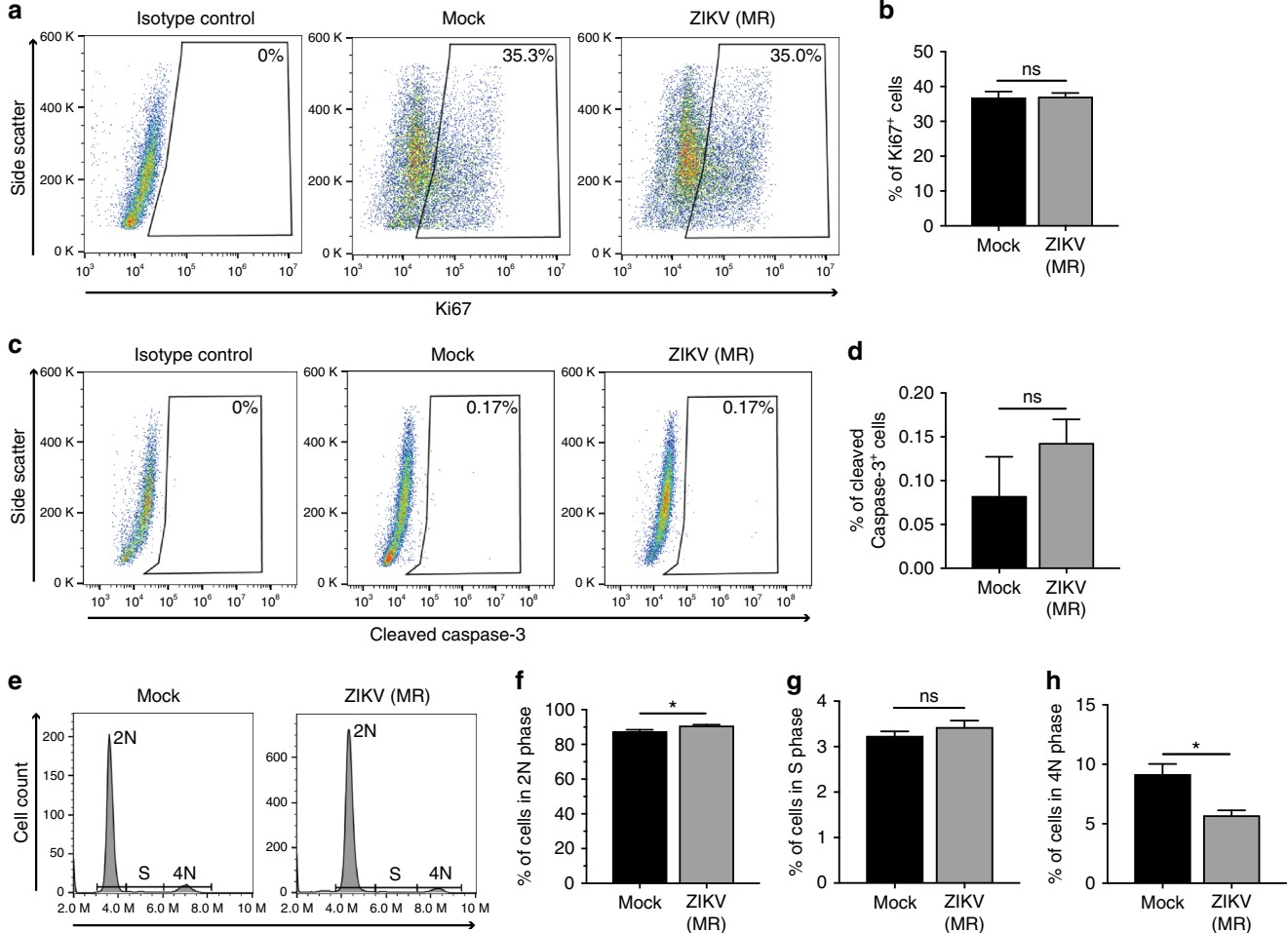

**Fig. 3** ZIKV induces minor effects on cell cycle but not on proliferation or survival. **a**, **b** Intracellular flow cytometry analysis (**a**) and quantification (*n* = 3) (**b**) of Ki67⁺ GCs 72 h after mock and ZIKV (MOI = 0.1 PFU per cell) infection. **c**, **d** Intracellular flow cytometry analysis (**c**) and quantification (*n* = 3) (**d**) of cleaved caspase-3⁺ GCs 72 h after mock and ZIKV (MOI = 0.1 PFU per cell) infection. **e–h** Cell cycle analysis with flow cytometry (**e**) and quantification of percentage of GCs in 2N phase (**f**), S phase (**g**), and 4N phase (**h**) 72 h after mock and ZIKV (MOI = 0.1 PFU per cell) infection. Statistical values are presented as mean ± s.e.m. and analyzed by two-sided *t* test and *p ≤ 0.05 or no significance (ns). Scale bar, 50 μm. ZIKV strain is MR 766

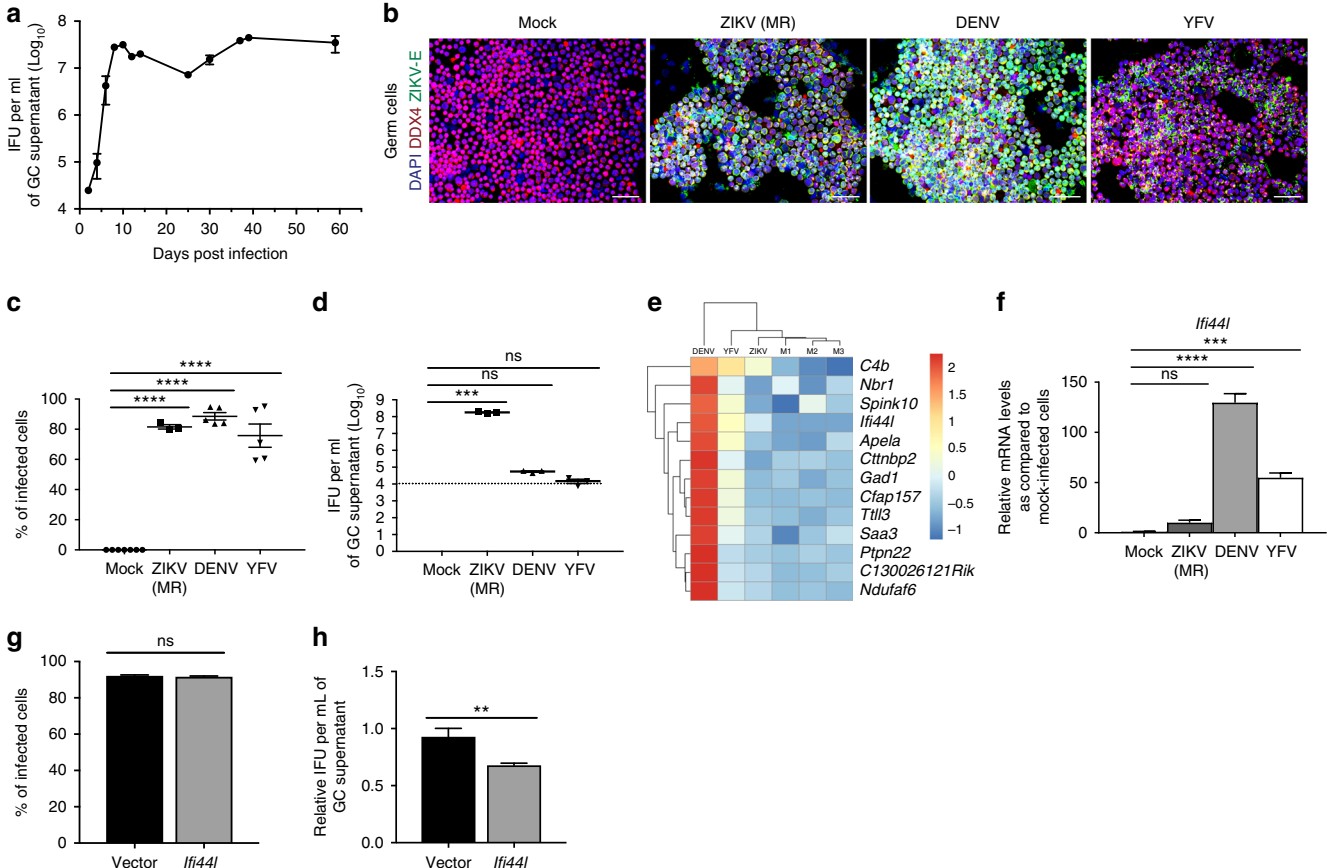

**Fig. 4** Male GCs propagate ZIKV due to reduced *Ifi44l* activation. **a** Assessment of long-term propagation of ZIKV in GCs up to 59 dpi with an intracellular flow cytometry-based Vero assay. **b–d** Immunostaining (**b**), the quantification of Fig. 4b ($n = 7$ for mock, $n = 3$ for ZIKV, and $n = 5$ for DENV and YFV) (**c**), and the quantification of infectious virus in the supernatant of male GCs infected by mock, ZIKV (MOI = 0.1 PFU per cell), DENV (MOI = 0.1 PFU per cell), and YFV (MOI = 0.1 PFU per cell) at 72 hpi (**d**). **e** Heatmap of differential expression pattern from RNA-seq data of mock-infected, ZIKV-infected, DENV-infected, and YFV-infected GC. **f** qRT-PCR analysis of *Ifi44l* mRNA levels in GCs infected with mock, ZIKV, DENV, and YFV at 72 hpi. **g** Percent of GC$^{Ifi44l}$ and GC$^{Vector}$ infected by ZIKV (MOI = 0.1 PFU per cell) at 72 hpi. **h** Infectivity analysis of supernatant from ZIKV-infected GC$^{Ifi44l}$ and GC$^{Vector}$ at 72 hpi with intracellular flow cytometry-based Vero assay. The relative number was normalized to IFU ml$^{-1}$ of supernatant of GC$^{Vector}$ at 72 hpi ($n = 14$ for GC$^{vector}$ and GC$^{Ifi44l}$). $n = 3$ for all experiments except where indicated; data are presented as mean ± s.e.m.; two-sided $t$ test for **g**, **h**; one-way ANOVA for **c**, **d**, **f**; **\*\***$p \leq 0.01$, **\*\*\***$p \leq 0.001$, **\*\*\*\***$p \leq 0.0001$, or no significance (ns). In **d**, the gray dashed line represents the limit of detection. ZIKV strain is MR 766 throughout. Scale bar 50 μm for **b**

**Male GCs propagate ZIKV due to reduced *Ifi44l* activation**. Given reports of long-term residual ZIKV in the semen of men with undetectable peripheral viremia[3], we evaluated the ability of GCs to support long-term ZIKV propagation in vitro. Remarkably, ZIKV-infected GCs continuously produced infectious virus for 59+ dpi with no decrease in production (Fig. 4a); similarly, GCs infected with ZIKV PRVABC59 continued to produce infectious virus through at least 34+ dpi (Supplementary Fig. 2a). Next, in order to evaluate whether infection of GCs is specific to ZIKV, we infected GCs with other flaviviruses. Interestingly, 88% and 75% of GCs were infected by dengue virus (DENV) and yellow fever virus (YFV), respectively (Fig. 4b, c and Supplementary Fig. 2b), suggesting that GCs are also susceptible to infection by other flaviviruses. However, DENV-infected and YFV-infected GCs did not efficiently produce infectious virus (Fig. 4d). RNA-sequencing (RNA-seq) was used to compare the gene expression profiles in mock-infected GCs with those infected with ZIKV, DENV, and YFV. Among the top 150 most-upregulated genes in DENV-infected (>10-fold) or YFV-infected (>4-fold) GCs, we selected genes that were upregulated in both DENV-infected and YFV-infected GCs but not in ZIKV-infected GCs. We found one ISG, *Ifi44l*, among those genes

(Fig. 4e). qRT-PCR analysis confirmed that *Ifi44l* was upregulated in DENV-infected and YFV-infected GCs by 130-fold and 55-fold, respectively, but not in ZIKV-infected GCs (Fig. 4f). To examine the effect of *Ifi44l* on ZIKV production, *Ifi44l* was overexpressed in GCs as confirmed by qRT-PCR (Supplementary Fig. 2c). No difference was detected in the percentage of ZIKV-infected cells between GC$^{Ifi44l}$ and GC$^{Vector}$ (Fig. 4g). Over-expression of *Ifi44l* in GCs resulted in a moderate reduction in the levels of infectious ZIKV in the supernatant (Fig. 4h). These data suggest a possible role for *Ifi44l* in limiting flavivirus production, and the ability to prevent *Ifi44l* induction may be linked to long-term production of high levels of ZIKV by infected GCs.

**Berberine chloride inhibits ZIKV infection**. Although multiple studies have identified anti-ZIKV compounds using chemical screening methods[14–22], none focused on the effect of the identified compound on ZIKV levels in the testis. Since GCs appear to be an important target of ZIKV infection in the testis, identification of a small molecule that prevents or limits ZIKV infection of GCs could potentially block sexual transmission of ZIKV. Although several compounds have been identified that reduce

ZIKV levels in murine models[18, 23–25], no prophylactic or therapeutic treatment is available to prevent the spread of ZIKV infection in humans. To identify compounds that could inhibit ZIKV infection in male GCs, we performed a small-scale drug repurposing screen using compounds previously reported to inhibit infection of other flaviviruses or target pathways believed to be important for ZIKV infection[26–34]. Of the 21 compounds tested, 7 significantly reduced the percentage of ZIKV+ GCs by more than 30% (Fig. 5a and Supplementary Fig. 3a–h). Berberine chloride (BC) (Fig. 5b), a plant-derived alkaloid, demonstrated

the strongest inhibitory effect on ZIKV infection with a half-maximal inhibitory concentration (IC$_{50}$) = 2 μM (Fig. 5c) and no appreciable toxicity at the concentrations used for its inhibitory effects (Supplementary Fig. 3d). Treatment with 30 μM BC led to >99% reduction in ZIKV infection (Fig. 5d), cellular ZIKV vRNA (Fig. 5e), negative-strand vRNA (Fig. 5f), and vRNA and infectious virus in the culture supernatant (Fig. 5g, h). BC also inhibited GC infection and generation of infectious ZIKV PRVABC59, and surprisingly, DENV and YFV infection of GCs as well, suggesting that this compound could potentially be a pan-

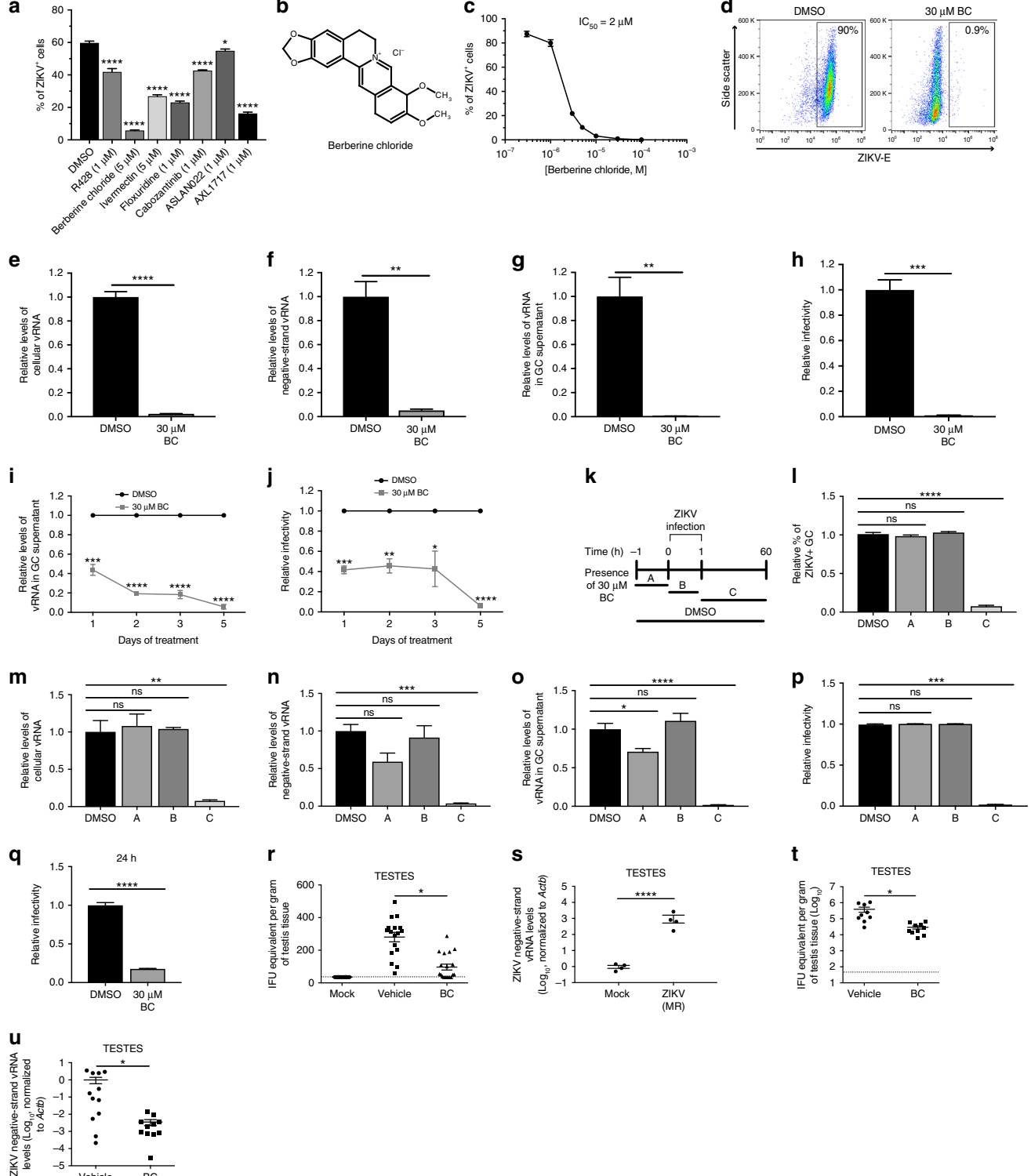

flavivirus inhibitor (Supplementary Fig. 3i, j). To determine BC's effect on established, long-term ZIKV MR 766-infected GCs, GCs at 70 dpi were treated with 30 µM BC. ZIKV vRNA and infectious virus were quantified in culture supernatants every 24 h by qRT-PCR (Fig. 5i) and flow cytometry (Fig. 5j). Five days of BC treatment was sufficient to reduce ZIKV levels by >94% as compared with untreated controls. Similarly, BC dramatically reduced ZIKV PRVABC59 vRNA levels in the GC supernatant (Supplementary Fig. 3k) and infectious virus production (Supplementary Fig. 3l) in long-term ZIKV PRVABC59-infected GCs.

Next, we performed a time-course study to further define the time point at which BC functions. GCs were either treated with dimethyl sulfoxide (DMSO) for the duration of the experiment (DMSO condition) or treated with BC for 1 h before infection (condition A), during the incubation for 1 h with ZIKV (condition B), or after ZIKV incubation for 60 h (condition C) (Fig. 5k). Samples were collected at 60 hpi for analysis. The percentage of infected cells (Fig. 5l), cellular and negative-strand ZIKV vRNA in GCs (Fig. 5m, n), supernatant vRNA (Fig. 5o), and infectious virus (Fig. 5p) were all significantly decreased in condition C, in which BC was only present after the infection occurred. Since decreased virus infectivity or replication can result in an overall reduction in vRNA levels observed in condition C, we preincubated ZIKV with 30 µM BC for 24 h, and then diluted this mixture 1:1000 and infected VCs. A 24 h pre-incubation with BC resulted in a >5-fold reduction in ZIKV infectivity (Fig. 5q), with a similar effect observed for the PRVABC59 strain (Supplementary Fig. 3m). To further confirm that the reduction of infectivity was not due to a direct effect of BC on VCs during infection, we added 30 nM BC (concentration of BC after 1:1000 dilution) directly to the medium during the ZIKV infection and did not find a reduction in infectivity of ZIKV on VCs (Supplementary Fig. 3n). We then tested whether BC can prevent ZIKV infection in wild-type mice. Pretreatment of mice with BC led to a 64% reduction of ZIKV vRNA levels in the testes at 6 dpi, indicating that BC is active as an anti-ZIKV agent in vivo (Fig. 5r). Furthermore, we investigated ZIKV levels in other tissues and bodily fluids. ZIKV vRNA was detected in the semen, brain, and seminal vesicles of some vehicle-treated mice, but no BC-treated mice had detectable ZIKV vRNA (Supplementary Fig. 3o–q). However, there was no decrease in ZIKV vRNA in the prostate of the BC-treated mice when compared with the vehicle-treated (Supplementary Fig. 3r). To evaluate if BC would effectively reduce ZIKV levels in tissue that has already been infected, we began BC treatment 1 day after intratesticular ZIKV infection, when replicating ZIKV negative-strand vRNA was already detected in the testis (Fig. 5s). We found that BC reduced ZIKV cellular vRNA and negative-strand vRNA in the testis by 89% and 99%, respectively (Fig. 5t, u). Together, the in vitro and in vivo data provide strong evidence to support that BC is effective in preventing and treating ZIKV infection of murine GC.

**ZIKV infects human male GCs and BC reduces ZIKV infection.** Since most studies have used murine models to study the effect of ZIKV on testis infection[7, 9, 35] and there are differences between mouse and human spermatogenesis[36], the extent to which the results from murine models can be translated to humans is unknown. Thus, we established primary human testicular tissue cultures to model human ZIKV infection. Human testicular biopsy samples were infected ex vivo for 1 h with an MOI of $1 \times 10^7$ IFU and harvested for immunostaining and qRT-PCR at 72 hpi. We found that human GCs are permissive for ZIKV infection as indicated by colocalization of ZIKV-E and DDX4 (Fig. 6a, b). Consistent with immunostaining results, we detected cellular and negative-strand vRNA in ZIKV-infected human testis tissue and vRNA in the culture supernatant (Fig. 6c–e), as well as infectious virus in the supernatant (Fig. 6f). Similar to the anti-ZIKV activity on mice, BC treatment reduced both cellular and negative-strand vRNA levels in ZIKV-infected human testis tissue (Fig. 6g, h) and ZIKV vRNA levels in the culture supernatant (Fig. 6i). Furthermore, BC reduced infectious virus in the supernatant from ZIKV-infected human testicular tissue when sampled at both 48 and 72 hpi (media were refreshed after each sampling) as determined by plaque assay titration (Fig. 6j). The reduction of ZIKV infectivity in the presence of BC was not a direct action of BC on Huh-7.5 cells since the addition of 3 µM BC (the concentration of BC used to titer virus produced in Fig. 6j) did not inhibit plaque formation when simply added during the adsorption of ZIKV to Huh-7.5 cells (Supplementary Fig. 4). These data demonstrate that human GCs are a target of ZIKV and provide strong evidence that BC may be effective in preventing human sexual transmission of ZIKV.

## Discussion

The evidence of ZIKV in semen and sperm after acute infection highlights the existence of a site in the testis where ZIKV can propagate in immunocompetent individuals. As conflicting results have been reported from multiple research groups[7–9, 35], the identity of the cell type(s) that host ZIKV in the testis has remained elusive. Most ZIKV studies have utilized mice that lack functional type I IFN signaling, yet most humans who are infected by ZIKV are immunocompetent. In our study, we established a model of ZIKV infection in immunocompetent mice to study ZIKV infection of the testis and demonstrated

---

**Fig. 5** BC inhibits ZIKV infection of murine male GCs in vitro and in vivo. **a** Hit compounds that inhibit ZIKV infection in GCs by more than 30%. **b** Chemical structure of BC. **c** Inhibition curve of BC. $IC_{50} = 2\,\mu M$. **d** Intracellular flow cytometry plots of GCs at 72 hpi with ZIKV (MOI = 0.1 PFU per cell) in the presence of 30 µM BC ($n = 6$). **e–h** qRT-PCR analysis of relative ZIKV cellular (**e**) and negative-strand (**f**) vRNA in GCs, vRNA in supernatant (**g**), and infectivity with Vero assay (**h**) in the presence of 30 µM BC. **i, j** BC significantly reduced ZIKV infection in long-term infected GC. Culture media containing BC were refreshed every 24 h (starting at 70 dpi) and assessed for relative ZIKV vRNA in the supernatant by qRT-PCR (**i**) and for infectivity with Vero assay (**j**). **k** Schematic representation of BC treatment time course for (**l–p**). **l** Relative percentage of ZIKV + GCs in the presence of 30 µM BC ($n = 9$), measured by intracellular flow cytometry. **m–p** qRT-PCR analysis of relative ZIKV cellular (**m**) and negative-strand (**n**) vRNA in GCs, vRNA in supernatant (**o**), and infectivity with Vero assay (**p**). **q** Infectivity of ZIKV after incubation with 30 µM BC for 24 h, measured by Vero assay. Samples were diluted 1:1,000. **r** qRT-PCR analysis of ZIKV vRNA in testes of BC-treated and vehicle-treated ZIKV-infected mice at 6 dpi ($n = 18$ for mock-infected, 18 for ZIKV-infected vehicle-treated, and 20 for ZIKV-infected BC treated). **s** qRT-PCR analysis of ZIKV negative-strand vRNA in testis of intratesticular mock-infected and ZIKV-infected mice at 1 dpi ($n = 10$ per group). **t, u** qRT-PCR analysis of ZIKV cellular (**t**) and negative-strand (**u**) vRNA in intratesticular ZIKV-infected mouse testis after BC or saline treatment 1 to 4 dpi ($n = 10$ per group). **e–j, l–q** Data are normalized to values obtained from DMSO-treated groups. $n = 3$ for all in vitro experiments, unless otherwise indicated. Statistical values are presented as mean ± s.e.m. and analyzed by two-sided $t$ test and one-way ANOVA, $*p \leq 0.05$, $**p \leq 0.01$, $***p \leq 0.001$, $****p \leq 0.0001$, or no significance (ns). ZIKV strain is MR 766 throughout

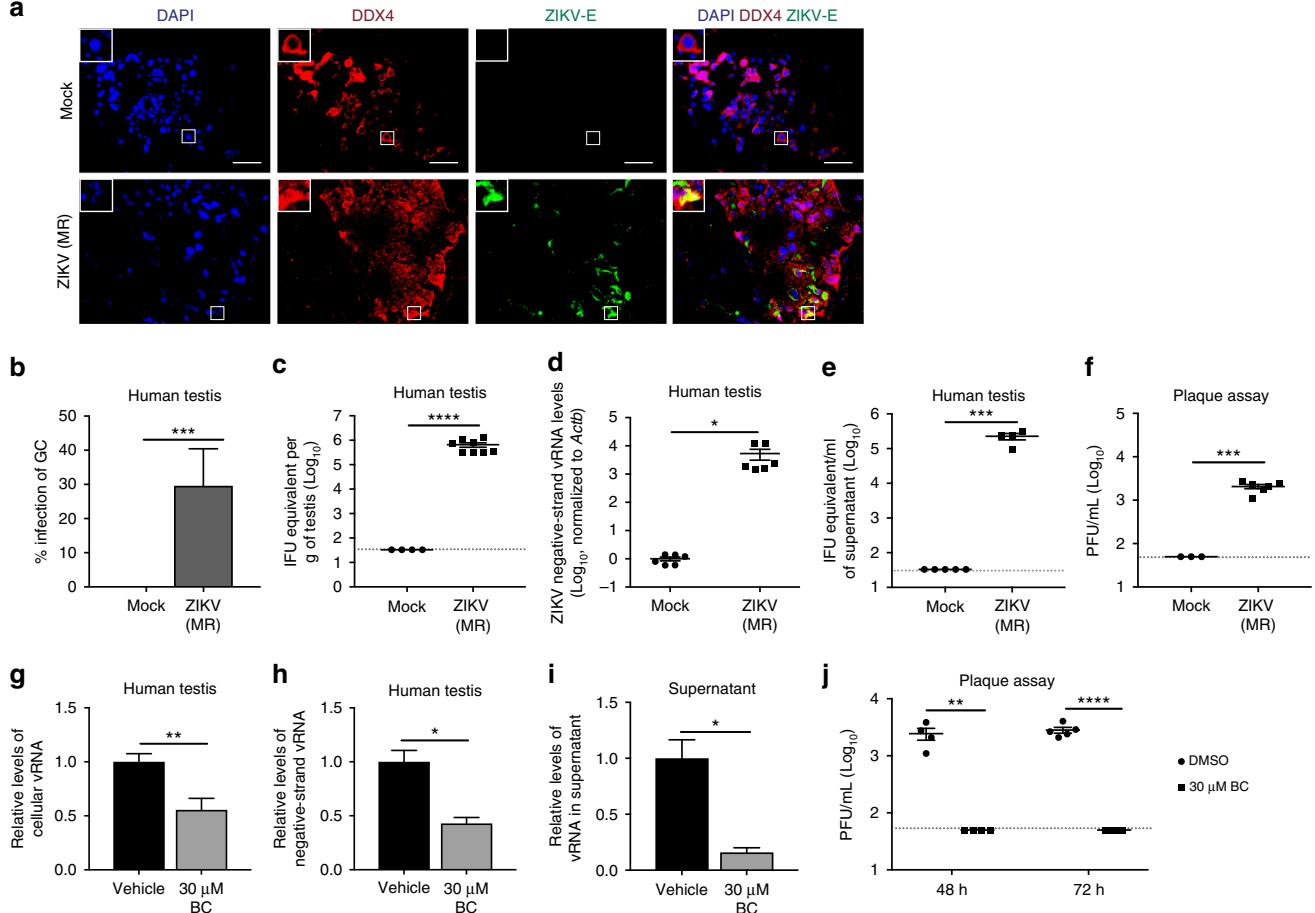

**Fig. 6** ZIKV infects human male GCs and BC reduces ZIKV infection. **a, b** Mock-infected and ZIKV-infected (IFU = 1 × 10⁷) human testis immunostaining with DDX4 and ZIKV-E (**a**) and quantification ($n = 4$ for mock-infected and 7 for ZIKV-infected) (**b**) at 72 hpi. **c–e** qRT-PCR analysis of ZIKV cellular vRNA in human testis ($n = 4$ for mock- and 8 for ZIKV-infected) (**c**), ZIKV negative-strand vRNA in human testis ($n = 6$ for mock-infected and ZIKV-infected) (**d**), and ZIKV vRNA in human testis culture supernatant ($n = 5$ for mock-infected and 4 for ZIKV-infected) (**e**) of mock-infected and ZIKV-infected (IFU = 1 × 10⁷) human testis samples at 72 hpi. **f** Infectivity of supernatant from mock-infected and ZIKV-infected human testis as assessed with plaque assay using Huh-7.5 cells ($n = 4$ for mock-infected and 6 for ZIKV-infected). **g–i** qRT-PCR analysis of ZIKV cellular vRNA in human testis (**g**), ZIKV negative-strand vRNA in human testis (**h**), and ZIKV vRNA in human testis culture supernatant (**i**) of BC-treated and DMSO-treated ZIKV-infected human testis samples. **j** Infectivity of supernatant from BC-treated and DMSO-treated ZIKV-infected human testis as assessed with plaque assay using Huh−7.5 cells ($n = 4$ for DMSO and BC for 48 h and $n = 6$ for DMSO and BC for 72 h). Statistical values are presented as mean ± s.e.m. and analyzed by two-sided $t$ test, $^*p \leq 0.05$, $^{**}p \leq 0.01$, $^{***}p \leq 0.001$, $^{****}p \leq 0.0001$. Scale bar, 50 μm. ZIKV strain is MR 766 throughout

significantly different susceptibility of SCs and MCs to ZIKV infection in wild-type and $Ifnar1^{-/-}$ mice, while GCs remained equally susceptible to ZIKV infection in both mouse models. This has important implications for future ZIKV studies as most ZIKV studies have focused on $Ifnar1^{-/-}$ mice or mice treated with IFN alpha/beta receptor subunit 1 blocking antibody. Since different cell types in the testis, and potentially other tissues, could show a differential response to IFN signaling, the use of $Ifnar1^{-/-}$ mice as a model may confound the identification of relevant cellular targets of ZIKV infection in immunocompetent individuals and host factors that regulate ZIKV infection and dissemination. Though GCs may exhibit the properties of a cell type that would be an ideal host to harbor ZIKV long term, there remains the possibility of ZIKV integration into the human genome. Although likely a very infrequent event, integrated flavivirus sequences have been identified in mosquito genomes[37, 38]. Such germline events could conceivably be deleterious, have adaptive value, and even provide a stable transgenerational repository of a flavivirus function or genome. ZIKV persistence in human GCs raises the intriguing possibility that ZIKV sequences may eventually become a passenger in the human germline.

## Methods

**Experimental design**. Required samples sizes were estimated using the results of preliminary experiments. Based on the methods for data collection in these experiments, blinding and randomization were not required. All samples and animals used in this study were included in the analysis.

**Cells and viruses**. Male GCs were derived from an adult 12-month-old wild-type mice on a mixed C57Bl6/Sv129 background (Regeneron Pharmaceuticals) and maintained on mitomycin-C-inactivated JK1 cells (Cell Biolabs, CBA-315), as previously described[39]. For propagation of GCs, JK1 feeder cells were plated on 6-well plates (BD Falcon, 62406-161) coated with 0.4% gelatin (Sigma-Aldrich, G1890-100G) at a density of 250,000 cells per well in Dulbecco's modified Eagle's medium (DMEM) (Cellgro, 45000-304) with 10% fetal bovine serum (Gibco, 10438026). After 18 h, JK1 cells were treated with mitomycin-C (10 μg ml⁻¹, Sigma-Aldrich, M4287-5X2MG). After 2.5 h, inactivated feeder cells were washed three times with DMEM. GCs were then plated on top of the inactivated JK1 feeder cells in GC culture media. GCs were passaged every 7 days using 0.05% trypsin/0.53 mM EDTA (Corning Cellgro, 25-051-CI). Culture medium for the GCs was StemPro-34 SFM (Gibco, 10640) supplemented with StemPro supplement, 1× (Gibco, 10641); D-glucose, 6 mg ml⁻¹ (Sigma-Aldrich, G8769-100ML); bovine serum albumin, 0.50% (Hyclone, G8769-100ML); MEM vitamin solution (Gibco, 11120-052); β-estradiol, 30 ng ml⁻¹ (Sigma-Aldrich, E2758-1G); progesterone, 60 ng ml⁻¹ (Calbiochem, 5341); fetal bovine serum, 1% (Gibco, 10438026); antibiotic–antimycotic solution (Gibco, 15240062); bovine holo-transferrin, 100 μg ml⁻¹ (Sigma-Aldrich, T1283-500MG); insulin, 25 μg ml⁻¹ (Sigma-Aldrich, I9278-5ML); human glial-derived

neurotrophic factor, 20 ng ml$^{-1}$ (Life, PHC7041); human basic fibroblast growth factor, 10 ng ml$^{-1}$ (Life, PHG0023); non-essential amino acid solution (Gibco, 11140-050); L-glutamine, 2 mM (Gibco, 45000-676); putrescine, 60 μM (Sigma-Aldrich, P7505-25G); sodium selenite, 30 nM (Sigma-Aldrich, 214485-5 G); pyruvic acid, 30 μg ml$^{-1}$ (Sigma-Aldrich, 107360-25 G); $d$(L)-lactic acid, 1 μg ml$^{-1}$ (Sigma-Aldrich, L1250-500ML); β-mercaptoethanol, 50 μM (Gibco, 21985-023); ascorbic acid, 100 mM (Sigma-Aldrich, A4544-25G); D-biotin, 10 μg ml$^{-1}$ (Sigma-Aldrich, B4639-500MG); and mouse epidermal growth factor, 20 ng ml$^{-1}$ (Life, PHG0313). The cells were maintained at 37 °C in an atmosphere of 5% carbon dioxide in air and fed three times per week. GC activity of cultured cells was checked periodically using transplantation assays[40]. VCs (ATCC CCL-91) were cultured at 37 °C in Eagle's Minimum Essential Medium (EMEM) supplemented with 10% fetal bovine serum (FBS). ZIKV MR 766 (NCBI GenBank accession No. KU720415) and PRVABC59 (NCBI accession No. KU501215) were obtained from Zeptometrix. All cell lines were tested quarterly for mycoplasma.

**Vero assay**. Determination of IFU ml$^{-1}$ for Vero assays was performed using an intracellular flow cytometry-based method as previously described[41]. Briefly, 80,000 VC cm$^{-2}$ were plated. After 12 h, GC supernatant was diluted at 1:100 or 1:1000 in 2% FBS/EMEM and dilution was added to the VCs for 1 h. Cells were then changed to virus-free media. After 24 h, cells were harvested for intracellular flow cytometry to quantify virus-infected cells.

**ZIKV plaque assay**. Huh-7.5 cells[42] (Laboratory of Dr. Charles Rice) were seeded at a density of $3.5 \times 10^5$ cells per well in 6-well plates (Costar) and cultured overnight in DMEM supplemented with 10% FBS and non-essential amino acid (Gibco). Cells were adsorbed with 10-fold serial dilutions of virus stock or cell culture supernatant diluted in Opti-MEM (Gibco) at 37 °C. After 1 h, cells were overlaid with 1.2% Avicel microcrystalline cellulose (FMC BioPolymer) prepared in DMEM with 10% FBS and incubated at 37 °C. At 4 dpi, cells were fixed in 7% formaldehyde and stained with 1.25% crystal violet solution prepared in 20% ethanol. Stain was removed and plates were washed with water and air-dried. Plaques were enumerated and viral titers were determined.

**Infection of human testicular tissue**. The studies have been approved by IRB committee of Weill Cornell Medical College. Human testicular tissue was obtained from biopsies of healthy male patients who had obstructive azoospermia due to vasectomy. Informed consent was obtained from subjects used in this study. The biopsied tissue was then infected with mock and ZIKV (MR 766) at an MOI of $1 \times 10^7$ IFU at 37 °C for 24 h on a shaker system in an incubator. The human in vitro fertilization tubal medium was removed from the infected tubules, tubules were washed two times, and the medium was replenished every 24 h until 72 hpi.

**Infection of cultured testicular cells**. Feeder subtraction was performed on the GCs, and infection was carried out in suspension with one million cells per ml in GC media at an MOI = 0.1 PFU per cell for 1 h. Cells were then washed two times before being plated on gelatin-coated plates at a density of 400,000 cells cm$^{-2}$. Infection continued for a total of either 60 h for the BC inhibition studies or 72 h for infection studies. SCs were isolated from 20-day-old C57BL/6J (The Jackson Laboratories) testes and cultured in vitro[43]. Cells were plated on Matrigel (BD Biosciences)-coated 12-well plates at $0.4 \times 10^6$ cells cm$^{-2}$ for infection. Cells were maintained in F12/DMEM (1:1) (Thermo Fisher Scientific) supplemented with growth factors and gentamicin as described[43]. LCs and MCs were isolated as previously described[8]. The aforementioned cells were plated at a density of 10,000 cells cm$^{-2}$ 24 h before infection. The following day, the cells were infected at an MOI = 0.1 PFU per cell for 1 h, the media containing the virus was removed, the cells were washed two times with virus-free media, and the infection was continued for 72 h. For IFNα studies, cells were preincubated with 1000 U ml$^{-1}$ for 18 h before infection.

**RNA-seq analysis**. The RNA-seq was performed as described previously[23]. Briefly, GCs at 72 hpi were collected for RNA-seq; samples included mock-infected, ZIKV-infected (MR 766), DENV-infected, and YFV-infected GCs (MOI = 0.1 PFU per cell). The RNA from each of these samples was extracted using the RNAeasy Mini Kit (Qiagen), the quality of RNA was validated using a bioanalyzer (Agilent), and cDNA libraries were created using the Illumina TruSeq Stranded mRNA Library Prep Kit and sequenced with single-end 50 bp on the Illumina HiSeq4000 instrument. STAR is used to align raw sequencing reads to the mouse GRCm38 reference genome[44], and Cufflinks was used to measure transcript abundances in fragments per kilobase of exon model per million mapped reads[45, 46]. To create the heatmaps displaying the differential gene expression patterns of the aforementioned samples, expression values (reads per kilobase million (RPKM)) were normalized per gene over all samples, the mean and standard deviation (STDEV) of expression over all samples were calculated for each gene, and the expression value was linearly transformed using the formula (RPKM-mean)/STDEV. The heatmap plot was generated using R heatmap package.

**IFNα stimulation of mouse testicular cells**. Testicular cells were isolated from 4-week-old C57BL/6 mice (The Jackson Laboratories) based on previously described

procedures[47]. The testes were decapsulated and incubated with 0.5 mg ml$^{-1}$ collagenase type 1 (Sigma-Aldrich) in F12/DMEM (Life Technologies) at room temperature (RT) for 15 min with gentle rocking. The suspensions were filtered through a 70-μm mesh to separate interstitial cells from seminiferous tubules. The seminiferous tubules were cut into small pieces and were incubated with 0.5 mg ml$^{-1}$ collagenase type 1 for 15 min at RT. The suspensions were filtered through a 70-μm mesh to remove chunks of undigested tissue. The primary cells were cultured in a 96-well tissue culture dish in F12/DMEM at 37 °C for 72 h before stimulation. Cells were starved in no FBS medium for 6 h, and then stimulated with 1000 U ml$^{-1}$ IFNα for 1 h. Cells were then fixed in 10% formalin and immunostained. To quantify the percentage of pSTAT1-responsive cells, MetaXpress was used to quantify number of immunostained cells that showed localization of pSTAT1 and the cell-type-specific markers (DDX4, VIM, CALB2, and αSMA).

**Long-term propagation of ZIKV in vitro**. The cultured GCs were plated at an initial density of 25,000 cells cm$^{-2}$ on mitomycin-C-inactivated feeder cells. The cultured GCs were infected with ZIKV MR 766 at an MOI = 0.1 PFU per cell or PRVABC59 at an MOI = 0.1 PFU per cell for 1 h. Each of the time points represent incubated media collected from a 48-h time period, and media were refreshed every 48 h. The sampled media at each time point was then used for virus titration by Vero assay, as mentioned previously, and for quantifying the level of ZIKV vRNA by qRT-PCR.

**Cell cycle analysis**. ZIKV-infected and mock-infected cultured GCs were digested, washed, and fixed with cold 70% ethanol and incubated overnight. The suspension was centrifuged and washed with phosphate-buffered saline (PBS). The resulting pellet was resuspended and incubated for 30 min in a solution of 1 mg ml$^{-1}$ of ribonuclease I in PBS. Propidium iodide was added to the solution for a final concentration of 10 μg ml$^{-1}$. Samples were interrogated on a BD Accuri C6 flow cytometer and analyzed using the FlowJo software.

**Drug repurposing candidate screen**. GCs were treated with compounds during ZIKV inoculation for 1 h. Cells were treated with compounds again after removal of ZIKV and were incubated for an additional 60 h. Compounds and concentrations used are indicated in Supplementary Table 1.

**MTS cytotoxicity assay**. Matrigel diluted 1:30 in DMEM was used to coat the wells of a 96-well plate for 1 h at 37 °C. GCs were plated at 50,000 cells per well in a 96-well plate and the compounds of the screen were added and incubated for a total of 60 h. At the end of the incubation period, 20 μl of solution from CellTiter 96® AQueous One Solution Cell Proliferation Assay (MTS) (Promega) was added for 4 h. The absorbance of each well was read at 490 nm.

**Immunofluorescence**. Cultured GCs were digested, washed, and resuspended in cold PBS containing 2% FBS and 1 mM EDTA, and cytospin was performed using 200,000 cells to pellet the cells on the slides at 800 rpm for 4 min. The GCs were then fixed with 10% formalin for 10 min. Other cultured cells were fixed for 20 min at RT. The cells were washed with PBS, blocked with blocking buffer containing 0.3% Triton X in 5% horse serum for 30 min at room temperature, and incubated with primary antibodies including rabbit anti-DDX4 (1:500, Abcam, ab13840), rabbit anti-SOX9 (1:500, Millipore, AB5535), rabbit anti-vimentin (1:500, Abcam, ab92547 [EPR3776]), rabbit anti-17β-HSD (1:100, Proteintech, 13415-1-AP), rabbit anti-α-SMA (1:500, Millipore, ABT1487), rabbit anti-CALB2 (1:250, Novus Biologicals, NBP1-32244), rabbit anti-DAZL (1:500, Abcam, ab34139), mouse and rabbit anti-flavivirus group antigen (ZIKV-E) (1:500, Absolute, Ab00230-2.0 and Ab00230-23.0 [D1-4G2-4-15 (4G2)]), rabbit anti-pSTAT1 (Tyr701) (1:300, Cell Signaling, 9167 [58D6]), mouse IgG (1:500, Jackson ImmunoResearch, 015-000-003), and rabbit IgG (1:200, Jackson ImmunoResearch, 011-000-003) at 4 °C overnight. Slides were washed with 0.3% Triton X in PBS, and incubated for 1 h at RT with secondary antibodies including mouse Alexa Fluor-488 (1:500, Thermo-Fisher, A-21202) and rabbit Alexa Fluor-594 (1:500, ThermoFisher, A-21207). Nuclei were counterstained with DAPI (1 μg ml$^{-1}$, ThermoFisher, D21490). Mouse testis tissue was fixed in 10% formalin at 4 °C, cryoprotected in 30% sucrose, and embedded in TissueTek® O.C.T. compound for cryosectioning. Serial 5-μm sections were taken on a Leica cryostat and stored at −20 °C. Testis samples were permeabilized and blocked by incubating samples in buffer containing 0.3% Triton X-100 and 5% horse serum in PBS for 1 h, followed by overnight incubation with primary antibodies at 4 °C. The next day, samples were washed with PBS and then incubated with secondary antibodies for 1 h at RT and protected from light. Samples were mounted in Prolong Gold (Life Technologies). Imaging was completed with an Olympus IX71 inverted microscope. For quantification of percentage of ZIKV-infected GCs, SCs, LCs and MCs in vivo, at least 3 testes and 10 tubules were analyzed in each group.

**Intracellular flow cytometry**. Sample preparation was done using the Foxp3/Transcription Factor Staining Buffer Set (eBioscience). Cells were digested with accutase, washed with PBS, fixed for 15 min in fixation/permeabilization solution, and washed with permeabilization buffer. Subsequently, primary antibodies,

including anti-flavivirus group antigen (ZIKV-E) (Absolute), rabbit anti-cleaved caspase-3 (BD Pharmingen, 559565 [C92-605]), and rabbit anti-Ki67 (Thermo-Fisher, RM-9106-S1 [SP6]), were added to cells for 30 min at RT in permeabilization buffer with 2% FBS. Cells were washed using permeabilization buffer and incubated with secondary antibody, including rabbit Alexa Fluor-488 (ThermoFisher, A-21206) and mouse Alexa Fluor-647 (ThermoFisher, A-31571), for 30 min at RT. Cells were washed and analyzed with a BD Accuri C6 flow cytometer and FlowJo software. All primary and secondary antibodies were used as 1:500 dilutions at RT.

**Real-time PCR**. Intracellular vRNA was extracted using the RNeasy Mini Kit. Cell supernatant vRNA was extracted using the QIAamp Viral RNA Mini Kit (Qiagen). Reverse transcription was carried out with random hexamer primers or strand-specific tagged primers of ZIKV plus β-actin (*ACTB*) gene-specific primer of human or mouse using High Capacity cDNA Reverse Transcription Kit (Thermo Fisher). Taqman quantitative PCR (qPCR) reactions were performed with PrimeTime Gene Expression 2× Master Mix (IDT DNA) and probes for ZIKV and human *ACTB* or mouse *Actb*. ZIKV expression level was then normalized to human *ACTB* or mouse *Actb*. Primer sets used in reverse transcription and qPCR and probes used in qPCR were listed in Supplementary Table 2. Amplification was performed in a LightCycler 480 II (Roche). Quantification of amplified qRT-PCR data was processed using the AbsQuant Fitpoints Function of the LightCycler(R) 480 SW 1.5 software. Log dilutions of the vRNA standard for the ZIKV MR 766 were similarly reverse transcribed and 10-fold serial dilutions were used during each qRT-PCR amplification. vRNA standard curve was used to determine the limit of detection of the assay.

**Mouse experiments**. All mouse experiments were performed in accordance with institutional regulations as set forth by the Weill Cornell Medical College Institutional Animal Care and Use Committee. The following 6–8-week-old mice were used for this study: CD-1 (Charles River Laboratories), C57BL/6J (The Jackson Laboratories), and *Ifnar1*$^{-/-}$ (The Jackson Laboratories). CD-1 mice were inoculated with $1 \times 10^8$ IFU diluted in EMEM through an intravenous route. For intratesticular injections of C57BL/6J mice, $1 \times 10^5$ IFU ZIKV MR 766 or mock was administered by inserting a needle vertically into the testis, virus was slowly released from the syringe as the needle was being withdrawn so that the entire testis was filled with the virus or medium. For *Ifnar1*$^{-/-}$ mice, $1 \times 10^2$ IFU was used for intratesticular injection. To study the effect of BC (Sigma) on viral titers in testes and semen, CD-1 mice were gavaged twice per day with 80 mg kg$^{-1}$ BC 2 days before or one day after virus inoculation until the end of the experiment. Tissue that was used for quantifying ZIKV vRNA was snap-frozen in liquid nitrogen immediately after harvest. Cellular vRNA from the tissue was extracted by homogenizing tissue in lysis buffer and then isolating RNA using RNeasy Mini Kit. To measure viral titer in serum, blood was collected from the tail vein, clotted at RT for 1 h, and centrifuged at $8000 \times g$ for 2 min to collect the serum. Semen was extracted from dissected cauda epididymis and vas deferens. ZIKV vRNA was isolated from semen and serum using QIAamp Viral RNA Mini Kit. ZIKV vRNA titers in mouse tissue and body fluid were determined by normalizing to vRNA isolated from virus stock of known viral titers. For quantification of ZIKV-infected testicular cells in mouse testis, tubules with >70% total infection were selected.

**Statistical analyses**. The number of independent biological samples, statistical tests, and specific *p* values are indicated in each figure and were analyzed by GraphPad Prism. Briefly, when two groups were compared, a *t* test was used to determine statistical significance, and when there are more than two sets of data, a one-way analysis of variance with a Tukey's test was used for post hoc multiple comparisons to determine statistical significance. *$p \le 0.05$, **$p \le 0.01$, ***$p \le 0.001$, and ****$p \le 0.0001$.

**Data availability**. The data that support the findings of this study are available from the corresponding author upon request. RNA-seq data is available in GEO database (accession number: GSE113177).

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

## Acknowledgements

S.C., a New York Stem Cell Foundation-Robertson Investigator, is funded by The New York Stem Cell Foundation (R-103), and a pilot award from the Center for Advanced Digestive Care of New York Presbyterian Hospital. C.L.R. is funded by the National Institute of General Medical Sciences of the National Institutes of Health under Award Number T32GM007739 and the National Institute of Diabetes and Digestive and Kidney Diseases of the National Institutes of Health under Award Number DP2DK098093. M.S. is supported by a grant from the Eunice Kennedy Shriver National Institute of Child Health and Human Development of National Institutes of Health (1DP2HD080352-01) and a grant from the New York State Stem Cell Science Board (C029156). C.Y.C. is supported by a grant from the Eunice Kennedy Shriver National Institute of Child Health and Human Development of the National Institutes of Health (R01 HD056034). L.A.M. is a New York Stem Cell Foundation-Druckenmiller Fellow. Generous donations to the Rockefeller University support ZIKV work of J.M.L., A.W.A., and C.M.R. We would like to thank Peter Marzuk, Eva Robinson, Jordan Robinson, and Jason Hoo-Fatt for assistance and support.

## Author contributions

A.W.A., G.J., J.J., and H.C. contributed equally to the paper. C.L.R., A.C.N.C., A.W.A., R.S.K., E.I.T., H.C., L.Z., C.M.R., C.Y.C., M.S., and S.C. designed the experiments; A.C.N.C. and G.J. performed all mouse experiments; V.L.D. and M.G. provided human testes samples. C.L.R., A.C.N.C., and G.J. performed in vitro experiments including immuno-fluorescence; C.L.R., G.J., and R.M.K. performed flow cytometry experiments, A.C.N.C. and G.J. performed all qRT-PCR; C.L.R., G.J., A.C.N.C., J.J., and H.R. performed cryo-sectioning including immunofluorescence; G.J., L.A.M., and E.D. performed tissue culture; A.W.A. performed the plaque assays; C.L.R. and A.C.N.C. performed all analyses; C.L.R., A.C.N.C., R.S.K., A.W.A., J.M.L., M.S., C.M.R., C.Y.C., M.S., and S.C. analyzed results and wrote the manuscript.
