## [Peer Review File · Nature Communications]

Reviewers' comments:

Reviewer #2 (Remarks to the Author):

The authors have acceptably address my major concerns by including new key pieces of data and revising the interpretations to more accurately reflect the findings.

Reviewer #3 (Remarks to the Author):

My original concerns have been addressed.

Reviewer #4 (Remarks to the Author):

The revised manuscript is much improved and is of interest. I have only minor concerns listed below.

First in Figure 1 the phospho-stat immunofluorescence and quantification is confusing. It looks like there are many more cells active in the plus interferon GC images. A higher magnification image should be shown to clearly discriminate the distinct cell types. How did the authors quantify the images? And y axis for m, and all other y axis, should be more clearly defined- not fold change, but fold change in what.

Second, the data presented that IFI44L is antiviral in GCs is weak at best. This should be toned down.

Third, cc 50s for drug must be shown in the cell types used here. Serial dilution of drug starting as high as you can go and quantifying cell number. A variety of assays can be used but not PI, actual quantification of cell number.

Lastly, the authors should cite the other papers that have done ZIKV drug screening.

Point-to-point response.

Reviewer #4 (Remarks to the Author):

The revised manuscript is much improved and is of interest. I have only minor concerns listed below.

First in Figure 1 the phospho-stat immunofluorescence and quantification is confusing. It looks like there are many more cells active in the plus interferon GC images. A higher magnification image should be shown to clearly discriminate the distinct cell types. How did the authors quantify the images?

Response: We have included a higher magnification image as an inset to the original image in Fig 11. For quantification of pSTAT1 responsive cells, we used MetaXpress to quantify immunostained cells that showed localization of pSTAT1 and the cell type specific markers (DDX4, VIM, CALB2, and α SMA). We have included quantification method in the method section at Page 13.

And y axis for m, and all other y axis, should be more clearly defined- not fold change, but fold change in what.

Response: We have modified all y-axis of “fold change” to include specific information regarding the fold change.

Second, the data presented that IFI44L is antiviral in GCs is weak at best. This should be toned down.

Response: We modified our text to “Overexpression of *Ifi44l* in GC resulted in a moderate reduction in the levels of infectious ZIKV in the supernatant (Fig. 2h)” at pages 5-6 to reflect our findings more accurately. We would be open to further discussion if reviewer recommends other appropriate descriptions.

Third, cc 50s for drug must be shown in the cell types used here. Serial dilution of drug starting as high as you can go and quantifying cell number. A variety of assays can be used but not PI, actual quantification of cell number.

Lastly, the authors should cite the other papers that have done ZIKV drug screening.

Response: We have performed an MTS assay (CellTiter 96® AQueous One Solution Cell Proliferation Assay) to assess the cytotoxicity of the 7 hit compounds as Supplementary Fig. 3b. Berberine Chloride showed minimal toxicity at the concentrations used for our ZIKV inhibition studies.

“Lastly, the authors should cite the other papers that have done ZIKV drug screening.”

Response: We have modified the main text on page 6 and included citation of other papers that have performed drug screening to identify anti-ZIKV compounds.

REVIEWERS' COMMENTS:

Reviewer #4 (Remarks to the Author):

In each figure panel it must state the drug concentrations used. In figure S3a and other panels in this figure it is not stated. In figure S3b the first three panels are not labeled as to which drugs they are. And if S3b is GCs, it should state this.

Point-to-point response.

Reviewer #4 (Remarks to the Author):

In each figure panel it must state the drug concentrations used. In figure S3a and other panels in this figure it is not stated. In figure S3b the first three panels are not labeled as to which drugs they are. And if S3b is GCs, it should state this.

Response: We have added the names of the drugs to Supplementary Figure 3b, which is now Supplementary Figure 3b-h. We have included the concentration of BC in all *in vitro* data in Supplementary Figure 3 and throughout the manuscript. We also added GC in the new Supplementary Figure 3b-h for clarification.